# Clear Cell Renal Cell Carcinoma, Diagnostic and Therapeutic Difficulties, Case Report and Literature Review

**DOI:** 10.3390/medicina58101329

**Published:** 2022-09-22

**Authors:** Weronika Stolpa, Angelika Stręk-Cholewińska, Agnieszka Mizia-Malarz

**Affiliations:** Department of Pediatrics, Upper Silesia Children’s Care Heatlh Centre, Medical University of Silesia, 16 Medykow Street, 40-752 Katowice, Poland

**Keywords:** clear cell renal cell carcinoma, sunitinib, nivolumab, nephrolithiasis, children

## Abstract

Nephroblastoma is the most common kidney tumour in children, constitutes about 85% of cases. Although renal cell carcinoma (RCC) is the second-most common kidney malignancy in children, it constitutes only about 2–6% of all cases. Currently, the basis of children’s RCC treatment is Umbrella Protocol of SIOP-RTSG, but, due to the rare diagnosis of this neoplasm in children, in difficult cases, treatment is based on the experience in adult patients with RCC. Nephrectomy improves prognosis and is usually performed at the first step of treatment. Acute kidney injury secondary to urolithiasis in a patient after nephrectomy due to RCC is a unique, very serious complication. Study design: We present a case of a 10-year-old boy with metastatic clear cell renal cell carcinoma (ccRCC) of the right kidney and an acute renal failure of the left kidney secondary to uric acid nephrolithiasis. Partial regression of the spread of ccRCC after 12.5-month treatment with sunitinib, followed by progression being observed and satisfactory effects and tolerance of nivolumab were observed later. Comorbidity of acute kidney injury during nephrolithiasis and ccRCC after nephrectomy in children is unique. Drugs used in the treatment clear cell carcinoma in adults (sunitinib and nivolumab), are also used in children with ccRCC.

## 1. Introduction

Although renal cell carcinoma (RCC) is formally the second most common kidney malignancy in children (Wilms tumour is diagnosed the most often in childhood), it is not diagnosed very often and constitutes approximately 2–6% of all kidney malignancies in children [1]. Clear cell renal cell carcinoma (ccRCC) is one histopathological type. Currently, the basis of children’s RCC treatment is Umbrella Protocol of SIOP-RTSG, but, due to the rare diagnosis of this neoplasm in children, in difficult cases, treatment is based on the experience in adult patients with RCC. Nephrectomy improves prognosis and is usually performed at the first stage of treatment [2,3].

Acute kidney injury secondary to nephrolithiasis in a patient after nephrectomy due to ccRCC is a rare, very serious complication.

Here we present a case of a 10-year-old boy with metastatic clear cell renal cell carcinoma of the right kidney and an acute renal failure of the left kidney secondary to uric acid nephrolithiasis.

### Case Report

A 10-year-old boy occasionally complaining of abdominal pain over the preceding 2 months was examined. He had a single fainting episode at school in December 2019. Baseline blood tests were performed which did not show abnormalities apart from a slightly elevated LDH (440.0 U/L; normal range up to 270 U/L). His urine was normal. Abdominal ultrasound showed a tumour of the right kidney. Abdominal computed tomography (CT) and magnetic resonance imaging (MRI) confirmed the presence of a tumour in the right kidney (with dimensions) sized 81 × 81 × 104 mm. The image additionally revealed renal vein infiltration, tumour capsule rupture, right hepatic lobe infiltration and suspected metastases to the right adrenal gland and the peritoneum. Furthermore, the ileocecal lymph nodes were imaged sized up to 15 mm (Figure 1). Neither the ultrasound nor the computed tomography showed urolithiasis.

The CT of the chest showed five undetermined nodules within the right lung, sized 5 mm and two nodules within the left lung, sized 4 mm (Figure 2). Furthermore, an enlarged hilar lymph node was shown on the left (sized 16 × 9 × 12 mm).

Due to the age of the child and the ambiguous radiological findings, inconsistent with Wilms tumour, a Tru-Cut biopsy of the right kidney tumour was performed.

Because of the long waiting time for the result of the histopathological examination (Christmas time), in view of the local tumour spread and the presence of distant metastases (pathological left hilar lymph node and numerous small nodular lesions within the lungs), the decision to start chemotherapy according to the SIOP Nephroblastoma December 2001 protocol was made. The parents signed an informed consent before starting the therapy. The first course of vincristine (VCR) + actinomycin D (ACTD) and second course of vincristine were administered accordingly. He did not have clinical and laboratories tumour lysis syndrome.

After 2 weeks of treatment, the ultrasound image of the tumour did not change. On the base of histopathology examination clear cell renal cell carcinoma (ccRCC) paediatric type with the following expression profile: CD10+, RCC+, CK+, EMA+/−, SMA−, desmin−, Melan A− was diagnosed.

Therefore, in order to remove the tumour, right nephrectomy was carried out and the postoperative period was uneventful. The histopathology examination of whole material confirmed the ccRCC. Tumour cells were also found in vessel lumen of the renal hilum, renal capsule, hilar adipose tissue and the renal collecting system. However, there were no tumour cells in the resected lymph nodes.

Postoperatively (day 9), the boy presented vomiting and abdominal pain which prompted urgent inpatient readmission. The laboratory tests showed elevated C-reactive protein (CRP) level, white blood count (WBC), as well as serum creatinine (1.79 mg/dL; normal range: 0.5–0.99 mg/dL), BUN (77.0 mg/dL; normal range: 10.0–38.0 mg/dL) and uric acid (7 mg/dL; normal range: 5.5–6.0 mg/dL) levels. The remaining parameters were within the normal ranges. The abdominal ultrasound revealed a large hematoma in the tumour bed. A surgical consultation was held which found no indications for reoperation.

Despite intensive hydration (1800 mL/12 h), the boy remained nauseous and vomiting, complaining of abdominal pain, with continuous elevated blood pressure and significantly impaired diuresis (140 mL/12 h), without haematuria and infection. Moreover, kidney function marker elevation was observed within 12 h: creatinine 2.4–3.28 mg/dL, BUN 77.0–87.0 mg/dL, potassium 4.7–5.9 mmol/L, uric acid 7.5–7.7 mg/dL. Follow-up abdominal ultrasound carried out 12 and 18 h later revealed a mild dilatation of the left renal collecting system and the left ureter (up to 4 mm). There was no evidence of urinary tract deposits or urine within the bladder. Therefore, the plain CT scan of the abdomen and pelvis was performed, which revealed two deposits: the first within the left ureterovesical junction and the second within the lower pole of the only preserved renal pelvis (Figure 3).

Since post renal mechanism of acute renal failure was confirmed, the paediatric urologist consultant decided to opt for endoscopic procedure, whereby a deposit was removed from the ureterovesical junction and a double J stent was inserted. The procedure resulted in immediate restoration of abundant diuresis, which enabled the conservative management to continue. Blood pressure and creatinine levels rapidly normalised following the procedure. The child began to take alkalizing agents and antibiotic prophylaxis of urinary tract infections.

Following a clinical improvement, the MRI of the head and bone scintigraphy (both unremarkable) were performed to inform staging. The treatment of invasive paediatric renal cell carcinoma T3, N1, M1 was commenced. According to the Umbrella Protocol of SIOP-RTSG and on the base of the opinion colleagues from the European Cooperative Study Group for Pediatric Rare Tumors (EXPeRT) junction and the second within the lower pole, we administered sunitinib for 28 days, followed by a 14-day break, start in February 2020. The dose for the first two courses was 15 mg/m^2^ BSA. A follow-up after 2 courses disease progression showed (Table 1). The parents did not agree to the resection of the lymph nodes and lesions from the lungs.

In view of this, sunitinib was escalated to 25 mg/m^2^ (for the 3rd to the 8th course). Subsequent imaging studies showed stable pulmonary lesions and a partial regression of hilar lymphadenopathy (CT of the chest once a 3 months).

The boy tolerated the treatment well. The only adverse effects observed were excessive weight gain secondary to hypothyroidism (he is on thyroid hormone replacement) and change in hair colour as a side effect of sunitinib.

After the 13th course we observed radiological progression of disease (CT chest and abdomen) [Table 1]. The parents agreed for the collection lymph nodes and lung metastases for histopathological and molecular analysis. The ccRCC diagnosis was confirmed. Second line treatment on the base of nivolumab (3 mg/kg every 2 weeks) was started. After the fourth course we noticed partial remission lung metastases and thorax lymph nodes (CT), but in the abdomen was local recurrence at the site of previous tumour (MRI). PET CT showed partial metabolic regression in all places. 

Laparotomy was performed with resection of the recurrence focus. Histopathology examination confirmed ccRCC. In molecular analysis cyclin-dependent kinase (CDK 12)—potential therapeutic target, was presence.

The boy is continuing nivolumab therapy with very good tolerance. Currently, he is after the 19th course. The last chest CT showed significant partial remission nodular lung changes, single, smaller paratracheal and parabronchial lymph nodes, stable lung hilus lymph nodes and without changes under the diaphragm (Table 1). The abdominal MRI and US are without changes. The left kidney appears normal in echo pattern, size and blood flow, with no signs of urinary retention.

The boy was surgically consulted again. He was second time disqualified from lung metastases and lymph nodes resection.

At the moment, 32 months after the diagnosis and 30,5 months following treatment commencement, the boy has remained in a good general condition. In view of good treatment tolerance, unchanged pulmonary lesions and partial regression of the left hilar and right tracheobronchial lymph node involvement, the decision to continue the therapy was made.

## 2. Discussion

Kidney tumours account for approximately 8% of all childhood cancers. Wilms tumour constitutes approximately 85% of those cases, being the predominant kidney malignancy in children up to the age of 3 [2,3]. This pattern, however, changes with age and renal cell carcinoma (RCC) constitutes half of all kidney tumours diagnosed in 10-year-olds. There is a known genetic link between vHL, Birt-Hogg-Dube or NF2 and higher incidence of RCC [2,3,4].

Clinical manifestation of nephroblastoma and renal cell carcinoma are similar. Sausville et al. [4] analysed clinical presentation of 132 patients with RCC, demonstrating that abdominal pain (43%), haematuria (37%) and abdominal rigidity on palpation (16%) were the most common symptoms.

The diagnosis of kidney tumours in children is largely based on imaging studies. This is relatively easy in Wilms tumour, whereby the characteristic image of the tumour, its infiltration into healthy kidney tissue and location within the kidney may be sufficient for the diagnosis and treatment commencement. In RCC the radiographic image is not specific. However, calcification is more common ultrasound finding in RCC than in nephroblastoma (14–24% vs. 5–8%) [5,6,7,8]. Similarly, lymphadenopathy (>1 cm in diameter) is also more common in RCC (30–40% of cases) than in Wilms tumour [4,9,10]. Out of 120 children with RCC enrolled in the Children’s Oncology Group study AREN03B2, 27.1% had local lymph node involvement determined in diagnostic imaging by size of >1 cm in diameter. Histology evaluation confirmed RCC infiltration in 60.8% of those cases [10]. The resection of the involved lymph nodes improves prognosis in RCC (1–3). Ushijama reported 5-year survival of <30% in patients with lymph node involvement [5]. Whereas positive lymph nodes were identified in diagnostic imaging in our patient, further histological assessment ruled out the presence of malignant cells. Metastases of RCC are usually found in lungs (40–65%), liver (35–57%), and—less commonly—bones (10–42%) [5].

Prognosis in RCC is linked to tumour staging and can range between complete remission (CR) rate of 92.5% in those diagnosed at stage 1, where nephrectomy is the only treatment, to 12.5% in those diagnosed at stage 4 [1,2,3,11,12,13]. There is a study where all patients diagnosed at stage 4 (11 of 24 enrolled) died throughout the follow up [1]. The analysis of 281 patients demonstrated that younger age was associated with larger tumour size and its higher stage (30% of patients had invasive RCC; 33.3% of patients presented with lymph node involvement) [14].

Primary treatment in localised RCC includes nephrectomy and local lymphadenectomy. In the metastatic RCC currently available treatment options include tyrosine kinase inhibitors, anti-angiogenic drugs (sunitinib, sorafenib), mTOR inhibitors, nivolumab or immunotherapy. Chemotherapy with gemcitabine, doxorubicin, oxaliplatin or irinotecan is less commonly used nowadays. There are no uniform treatment guidelines for unresectable and metastatic RCC as the condition is particularly rare and treatment response data are scarce [2,3,12,13].

Our patient was initially treated with two courses of preoperative chemotherapy for Wilms tumour (VCR+ACTD and VCR at one week interval) which did not alter the ultrasound image of tumour and local lymph nodes. There are published cases of ineffective preoperative chemotherapy for Wilms tumour administered to children with RCC due to initial misdiagnosis as nephroblastoma is a more common paediatric malignancy [15].

There are reports supporting the use of anti-angiogenic agents to prolong the time to progression (TTP) in children with RCC. Sunitinib therapy was associated with TTP increase by 6 months to 7.75 months [16,17]. In a study carried out in 11 patients with stage 4 RCC, anti-angiogenic therapy was used as the first-line treatment [1]. The longest mean TTP was achieved with axitinib (*n* = 2; TTP = 7.8 months; range 5.5–10 months), followed by sunitinib (*n* = 6; TTP = 4.7 months; range 0.3–12 months). The discussed patient has been treated with sunitinib for a year. Partial regression of the hilar lymph node and a stable lung nodule pattern have been achieved. The tumour bed and subdiaphragmatic lymph nodes appear normal.

Urolithiasis is a condition of adulthood, with only approx. 1% of cases being individuals below 18 years of age [18]. Bad dietary habits can promote urolithiasis in genetically predisposed patients. Our patient with a history of right nephrectomy due to RCC, without tumour lysis syndrome after preoperative chemotherapy, had a low fluid intake and there were further dietary errors identified on assessment. This led to vomiting, subsequent dehydration increasing the specific gravity of urine. Considering the prerenal acute kidney injury of his only kidney, he was initially rehydrated, with no effect.

The diagnosis of urolithiasis in children is uncommon, which may be attributable to spontaneous stone expulsion seen in 41–63% of affected children. Spontaneous stone expulsion is more common in children than in adults and particularly involves cases with stones sized below 5 mm [5,18]. In urolithiasis, the abnormal composition of substances excreted with urine leads to deposit formation. Calcified deposits are usually well visible on X-rays, ultrasound and CT scans. Non-calcified stones (e.g., urine) in turn, are invisible on X-rays (non-shadowing deposits), hence difficult to assess. This type of lithiasis is called “silent” urolithiasis [19]. Non-calcified deposits can be imaged in computed tomography. The ultrasound image of non-calcified deposits is indistinguishable from other types of deposits because, like calcified deposits, they can also produce an acoustic shadow. In our patient, the resulting non-calcified deposits, most likely uric acid deposits (precise urine diagnostics to determine the type of urolithiasis has not been performed yet), led to postrenal acute kidney injury. Subsequent ultrasound assessments did not indicate urolithiasis, which could have been attributable to the deposit location at the left ureteral orifice and the absence of an acoustic window, which would be a filled bladder if the contralateral kidney was preserved and functional. Surprisingly, there was only minimal left pyelocaliceal and ureteral dilatation. That is why the computed tomography was needed to ultimately make a diagnosis of nephrolithiasis, upon which appropriate treatment could have been commenced. The alternative diagnostic management, which we did not opt for, would include saline administration into the bladder to artificially create an acoustic window enabling the assessment of the lower ureteral segment.

We present unique case as comorbidity of acute kidney injury during nephrolithiasis and ccRCC after nephrectomy. There are also few reports of partial regression of the spread of ccRCC after 12.5-month treatment with sunitinib, followed by progression and satisfactory effects and tolerance of nivolumab.

This case report adds to the body of evidence informing clinical practice and supporting drugs, such as sunitinib and nivolumab, in treatment of children with ccRCC.

## 3. Conclusions

1. Drugs used in the treatment clear cell carcinoma in adults (sunitinib and nivolumab), are also used in children with ccRCC. There is a need to exchange experiences on the treatment of rare cancers in children, incl. through publications to develop standards of conduct.

2. Comorbidity of acute kidney injury during nephrolithiasis of the only kidney is unique. Patients after unilateral nephrectomy should take special care of the urinary tract.

## Figures and Tables

**Figure 1 medicina-58-01329-f001:**
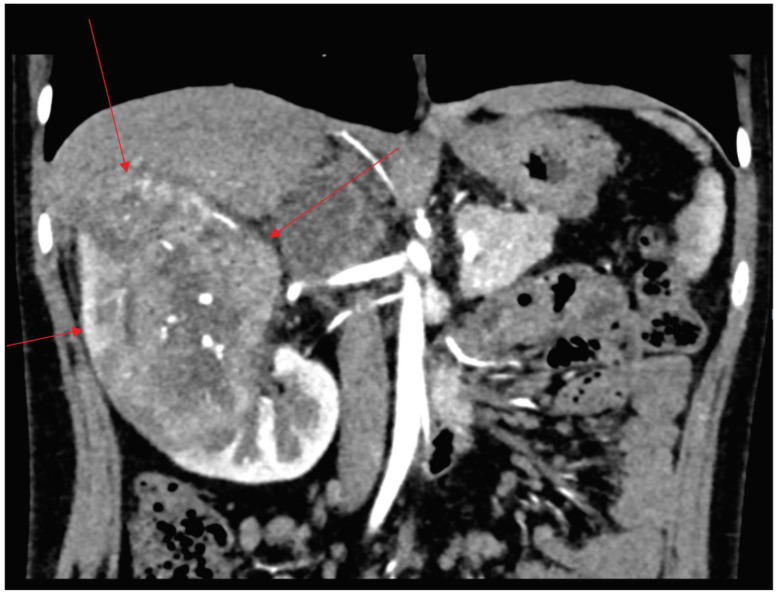
The abdominal computed tomography (CT) with visible the right kidney tumour (marked with an arrow), capsule rupture and right hepatic lobe infiltration.

**Figure 2 medicina-58-01329-f002:**
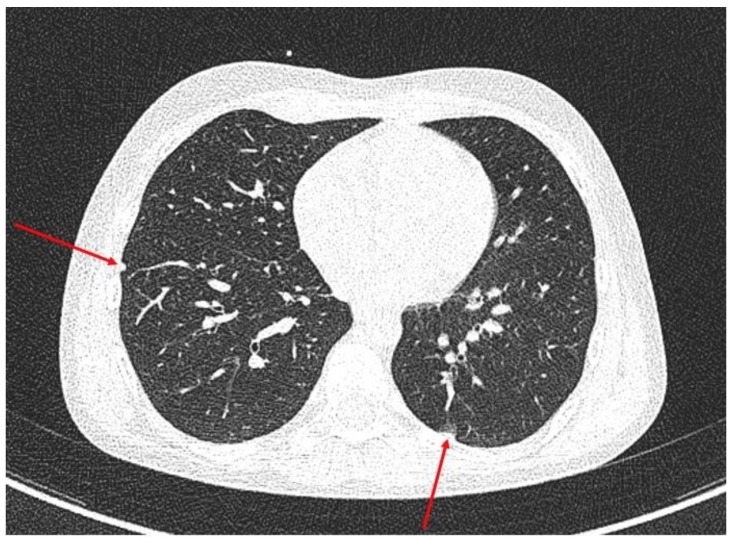
The CT of the chest with nodules within the right and left lungs (marked with an arrows).

**Figure 3 medicina-58-01329-f003:**
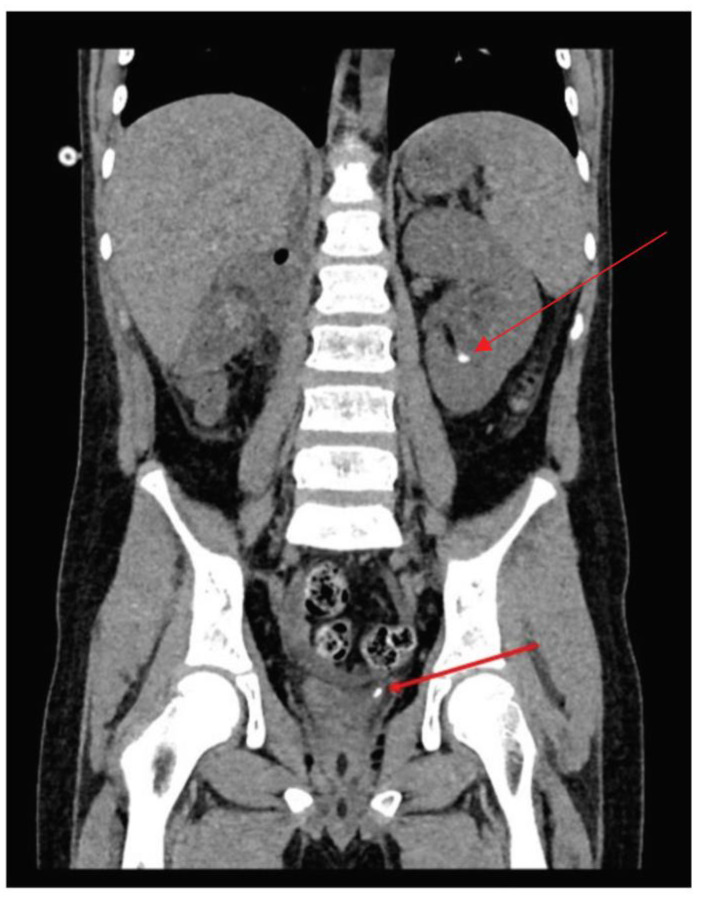
The CT of the abdomen and pelvis with 2 deposits: one within the left ureterovesical junction (lower arrow) and another one within the lower pole of the only preserved renal pelvis (upper arrow).

**Table 1 medicina-58-01329-t001:** Imaging tests (CT, MRI and PET CT) during therapy.

	CT Chest	CT/MRI Abdomen	PET CT
Diagnosis	Right lung: 5 nodules (5 × 5 mm) Left lung: 2 nodules (4 × 5 mm) Left hilus: 2 lymph nodes (max size 16 mm).	CT and MRI: a tumour in the right kidney (81 × 81 × 104 mm), renal vein infiltration, tumour capsule rupture, right hepatic lobe infiltration, suspected metastases to the right adrenal gland and the peritoneum. ileocecal lymph node (15 × 5 mm)	confirmed active disease process in the same places as in CT/MRI
Follow up after kidney tumor resection and 2 cycles sunitinib 15 mg/m^2^	unchanged size of previously described pulmonary lesions, but Right lung: a new single nodular lesion (10th segment) Right hilus: new 2 lymph nodes (10 × 10 and 20 × 11 mm)	Ileocecal lymph node PR (12 × 5 mm)	confirmed active disease process in the same places as in CT/MRI
Follow up after 8 cycles sunitinib 25 mg/m^2^	Right lung: 3 nodules (3 × 4 mm) Left lung: 2 nodules (3 × 3 mm) Left hilus: 2 lymph nodes max size 10 mm Right hilus: 2 lymph nodes (10 × 5 and 15 × 11 mm)	Ileocecal lymph node SD (13 × 5 mm)	
Follow up after 13 cycles sunitinib 25 mg/m^2^	Right lung: PD 3 nodules (max size 9 mm) Left lung: PD 2 nodules (max size 4 × 6 mm) Left hilus: SD Right hilus: SD	Ileocecal lymph node PD (12 × 19 mm)	
Follow up after 4 cycles of nivolumab 3 mg/kg mth	Right lung: PR 3 nodules (max size 5 mm) Left lung: PR 2 nodules (max 3 mm)	Ileocecal lymph nodes SD (13 × 19 mm)	Partial metabolic regression
Follow up after 18 cycles nivolumab 3 mg/kg mth	Right lung: PR 3 nodules (max 3 mm) Left lung SD Left, right hilus lymph nodes: SD	CR	Complete metabolic regression all showed places

Legend: PR-partial remission, SD-stable disease, CR-complete remission, PD-progression disease.

## Data Availability

Department of Oncology, Hematology and Chemotherapy Katowice, Poland.

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
