# Peer review of "Clear Cell Renal Cell Carcinoma, Diagnostic and Therapeutic Difficulties, Case Report and Literature Review"

_medicina, 2022, doi:10.3390/medicina58101329_

Round 1

Reviewer 1 Report (Previous Reviewer 2)

Mizia-Malarz et al. describe the medical history of a 10-year old boy with metastatic ccRCC and urolithiasis in form of a case report.

The report has a clear structure, and is easy to follow, I have only a few points to address:

·         Figure 1 does not show any arrows although described in the Figure legend and Figure 3 only contains one arrow instead of two…

·         The entire manuscript still needs to be checked for grammar and spelling!

Author Response

Response to Reviewer 1

We truly appreciate the effort you have made to read and assess our paper:

Clear cell renal cell carcinoma of the right kidney, nephrolithiasis and acute kidney injury of the left kidney - diagnostic and therapeutic difficulties

Weronika Stolpa, Angelika Stręk-Cholewińska, Agnieszka Mizia-Malarz,

We have revised and updated our manuscript.

  1. We changed the title of the manuscript: Clear cell renal cell carcinoma, diagnostic and therapeutic difficulties, case report and literature review.
  2. We corrected the arrows on the pictures.
  1.              The manuscript has been checked by English speaking colleague.

We hope that your opinion will be positive now.

Yours sincerely, Agnieszka Mizia-Malarz

Reviewer 2 Report (Previous Reviewer 1)

This is a further updated manuscript about a case report of a boy with ccRCC and nephrolithiasis. The paper has improved comapred to the versions before.

Nevertheless, english needs further improvement as can already be seen in the 2nd sentence of the abstract:'Although renal cell carcinoma (RCC) clear cell cc is the most common kidney malignancy in children, it constitutes only about 2-6% of all cases.' As the authors do kn ow, nephroblastoma is the most common kidney cancer in childhood. In addition, cc needs to be written as (cc). In several parts of the text, also in the abstract the followong is written: 'clear cell clear cell renal cell carcinoma' 'clear cell' needs to be written only once.

All other comments of the reviewers are addressed.

Author Response

Response to Reviewer 2

We truly appreciate the effort you have made to read and assess our paper:

Clear cell renal cell carcinoma of the right kidney, nephrolithiasis and acute kidney injury of the left kidney - diagnostic and therapeutic difficulties

Weronika Stolpa, Angelika Stręk-Cholewińska, Agnieszka Mizia-Malarz,

We have revised and updated our manuscript.

  1. We changed the title of the manuscript: Clear cell renal cell carcinoma, diagnostic and therapeutic difficulties, case report and literature review.
  1. The manuscript has been checked by a native speaking colleague.                      

We hope that your opinion will be positive now.               

Yours sincerely, Agnieszka Mizia-Malarz

This manuscript is a resubmission of an earlier submission. The following is a list of the peer review reports and author responses from that submission.

Round 1

Reviewer 1 Report

The paper describes a case report of RCC and urolithiasis in a 10 year old boy.

There are some issues that need to be addressed:

  1. RCC accounts for 2-4% of renal tumors in childhood and is only in older children (> 10 years the second most common renal tumor). In addition ccRCC is very rare in childhood. It is the most common RCC in adulthood.
  2. Ther are already guidelines available dealing with RCC (diagnosis and treatment). See e.g. the UMBRELLA Protocol of SIOP-RTS
  3. Urolithiasis is associated with an increased RCC risk in adults (doi: 10.1038/s41416-018-0356-7; doi: 10.1016/j.canep.2011.09.006) This needs to be discussed.
  4. Was informed consent given by the parents for preoperative chemotherapy and is the SIOP 2001 procol approven by the local Ethical committee?
  5. Did the patient had hematuria at diagnosis of RCC and/or at diagnosis of urolithiasis
  6. How was uric acid urolithiasis proven and how were other resons ruled out like calcium stones? was an infection of the urine ruled out?
  7. What happened with the high blood pressure after normalisation of renal failure?
  8. Why is no removal of lung metastasis done. It is unlikely that the child will survive if a complete remission cannot be achieved?
  9. If in the discussion an overview about RCC is given then major literauture is missing (doi: 10.1002/ijc.33476; doi: 10.3390/cancers12071776; doi: 10.1016/j.urolonc.2015.06.009; doi: 10.1002/cncr.22346)
  10. English language must be improved by an English speaking person

Author Response

Response to Reviewer 1

We truly appreciate the effort you have made to read and assess our paper:

Clear cell renal cell carcinoma of the right kidney, nephrolithiasis and acute kidney injury of the left kidney - diagnostic and therapeutic difficulties

Agnieszka Mizia-Malarz, Weronika Stolpa, Angelika Stręk-Cholewińska, Zbigniew Olczak, Katarzyna Gruszczyńska

Your suggestions have been of high value.

Point 1.

Ther are already guidelines available dealing with RCC (diagnosis and treatment). See e.g. the UMBRELLA Protocol of SIOP-RTS.

Answer 1.

I know the UMBRELLA Protocol of SIOP-RTS but part for RCC is short and without therapy details. Our case is meant to be a voice in the discussion of RCC in children.

Point 2.

Urolithiasis is associated with an increased RCC risk in adults (doi: 10.1038/s41416-018-0356-7; doi: 10.1016/j.canep.2011.09.006) This needs to be discussed.

Answer 2.

According to articles, urolithiasis is associated with an increased RCC (papillary type)  risk in adults. We don’t have information about this situation in children.

Point 3.

Was informed consent given by the parents for preoperative chemotherapy and is the SIOP 2001 procol approven by the local Ethical committee?

Answer 3.

The parents gave us the informed consent agreement before preoperative chemotherapy. The SIOP 2001 protocol by the local Ethical committee was approved a few years ago.

Point 4.

Did the patient had hematuria at diagnosis of RCC and/or at diagnosis of urolithiasis

Answer 4.

This boy didn’t have hematuria either at diagnosis of RCC or at diagnosis of urolithiasis.

Point 5.

How was uric acid urolithiasis proven and how were other reasons ruled out like calcium stones? was an infection of the urine ruled out?

Answer 5.

He didn’t have infection. We haven’t known what types of stones they were. We can only suspect on the base of USG/CT pictures.

Point 6.

What happened with the high blood pressure after normalisation of renal failure?

Answer 6.

After normalization of renal failure his blood pressure  normalized.

Point 7.

Why is no removal of lung metastasis done. It is unlikely that the child will survive if a complete remission cannot be achieved?

Answer 7.

According to surgeons opinion removing lung metastases and pulmonum hilus lymph nodes is very difficult…

Point 8.

If in the discussion an overview about RCC is given then major literauture is missing (doi: 10.1002/ijc.33476; doi: 10.3390/cancers12071776; doi: 10.1016/j.urolonc.2015.06.009; doi: 10.1002/cncr.22346).

Answer 8.

Thank you so much for new literature.

Point 9.

English language must be improved by an English speaking person

Answer 9.

This manuscript was corrected by native speaker.

I would like to thank you again for the valuable remarks. I hope the final review of this paper will be positive.

                                                                                   Yours sincerely, Agnieszka Mizia-Malarz

Reviewer 2 Report

Mizia-Malarz et al. describe in a case report the medical history of a 10-year old boy with ccRCC and urolithiasis. analyzed twelve human kidney specimen from tumor nephrectomies by histology and immunohistochemistry. They describe expression patterns of inflammatory and fibrotic markers.

The report has a clear structure, and is easy to follow, I have only a few points to address:

  • How will the sunitinib therapy be continued? Following the 3/1 scheme?
  • Kidney stones are not very common in children. Was the urine analyzed to identify the type of kidney stone? Can genetic predisposition for nephrolithiasis (and thereby recurrence) be excluded? Does the patient now follow dietary restrictions?
  • The entire manuscript should be checked for grammar and spelling.

Author Response

Response to Reviewer 2

We truly appreciate the effort you have made to read and assess our paper:

Clear cell renal cell carcinoma of the right kidney, nephrolithiasis and acute kidney injury of the left kidney - diagnostic and therapeutic difficulties

Agnieszka Mizia-Malarz, Weronika Stolpa, Angelika StrÄ™k-CholewiÅ„ska, Zbigniew   Olczak, Katarzyna GruszczyÅ„ska

Your suggestions have been of high value.

Point 1.

How will the sunitinib therapy be continued? Following the 3/1 scheme?

Answer 1.

It is mistake in manuscript.., he sunitinib was and is still used following the 4/2 scheme (4 weeks therapy, 2 weeks off schedule). It is corrected.

Point 2.

Kidney stones are not very common in children. Was the urine analyzed to identify the type of kidney stone?

Answer 2.

We didn’t analyze urine for the type kidney stones.

Point 3.

Can genetic predisposition for nephrolithiasis (and thereby recurrence) be excluded?

Answer 3.

He and his family are before genetic procedure for genetic  cancers and nephrolithiasis predisposition.

Point 4.

Does the patient now follow dietary restrictions?

Answer 4.

He has had dietary restrictions.

Point 5.

The entire manuscript should be checked for grammar and spelling.

Answer 5.

This manuscript was corrected by native speaker.

I would like to thank you again for the valuable remarks. I hope the final review of this paper will be positive.

                                                                                   Yours sincerely, Agnieszka Mizia-Malarz

Round 2

Reviewer 1 Report

This is an updated version of the article dealing with RCC, nephrolithiasis and renal failure in a child.

It is not true that there are not treatment recommendations for RCC in childhood in teh UMBRELLA protocol of SIOP-RTSG. This needs to be corrected.

The literature provided in the first review is not used in the discussion. None of these articles is taken into consideration.

Besides other treatment options one need to discuss why lung metastasis were not operated.

If it is unclear which types of stones caused the renal failure thsi needs to be discussed.

Author Response

Response to Reviewer 1

We truly appreciate the effort you have made to read and assess our paper:

Clear cell renal cell carcinoma of the right kidney, nephrolithiasis and acute kidney injury of the left kidney - diagnostic and therapeutic difficulties

Agnieszka Mizia-Malarz, Weronika Stolpa, Angelika Stręk-Cholewińska, Zbigniew Olczak, Katarzyna Gruszczyńska

Your suggestions have been of high value second time.

Point 1.

It is not true that there are not treatment recommendations for RCC in childhood in teh UMBRELLA protocol of SIOP-RTSG. This needs to be corrected.

Answer 1.

I agree with you..I corrected my manuscript.

Point 2.

The literature provided in the first review is not used in the discussion. None of these articles is taken into consideration.

Answer 2.

The submitted articles are very valuable to my knowledge, future treatment of this patient and other patients. Thank you so much. I have supplemented manuscript with these articles.

Point 3.

Besides other treatment options one need to discuss why lung metastasis were not operated.

Answer 3.

This boy was disqualified from of lung metastases resection and lymphadectomy twice times.

Point 4.

If it is unclear which types of stones caused the renal failure this needs to be discussed.

Answer 4.

We didn’t analyze urine for the type stones. I improved it in the manuscript.

I would like to thank you again for the valuable remarks. I hope the final review of this paper will be positive.

                                                                                   Yours sincerely, Agnieszka Mizia-Malarz

Round 3

Reviewer 1 Report

This is a second update of the paper. There is progress, despite not all mentioned literature in the review is taken iis discussed.